# Multivariate Convolutional Sparse Coding for Electromagnetic Brain Signals

**Tom Dupré La Tour**[*1], **Thomas Moreau**[*2], **Mainak Jas**[1], **Alexandre Gramfort**[2]

1: LTCI, Télécom ParisTech, Université Paris-Saclay, Paris, France
2: INRIA, Université Paris-Saclay, Saclay, France
*: Both authors contributed equally.

## Abstract

Frequency-specific patterns of neural activity are traditionally interpreted as sustained rhythmic oscillations, and related to cognitive mechanisms such as attention, high level visual processing or motor control. While alpha waves (8–12 Hz) are known to closely resemble short sinusoids, and thus are revealed by Fourier analysis or wavelet transforms, there is an evolving debate that electromagnetic neural signals are composed of more complex waveforms that cannot be analyzed by linear filters and traditional signal representations. In this paper, we propose to learn dedicated representations of such recordings using a multivariate convolutional sparse coding (CSC) algorithm. Applied to electroencephalography (EEG) or magnetoencephalography (MEG) data, this method is able to learn not only prototypical temporal waveforms, but also associated spatial patterns so their origin can be localized in the brain. Our algorithm is based on alternated minimization and a greedy coordinate descent solver that leads to state-of-the-art running time on long time series. To demonstrate the implications of this method, we apply it to MEG data and show that it is able to recover biological artifacts. More remarkably, our approach also reveals the presence of non-sinusoidal mu-shaped patterns, along with their topographic maps related to the somatosensory cortex.

## 1 Introduction

Neural activity recorded via measurements of the electrical potential over the scalp by electroencephalography (EEG), or magnetic fields by magnetoencephalography (MEG), can be used to investigate human cognitive processes and certain pathologies. Such recordings consist of dozens to hundreds of simultaneously recorded signals, for durations going from minutes to hours. In order to describe and quantify neural activity in such multi-gigabyte data, it is classical to decompose the signal in predefined representations such as the Fourier or wavelet bases. It leads to canonical frequency bands such as theta (4–8 Hz), alpha (8–12 Hz), or beta (15–30 Hz) (Buzsaki, 2006), in which signal power can be quantified. While such linear analyses have had significant impact in neuroscience, there is now a debate regarding whether neural activity consists more of transient bursts of isolated events rather than rhythmically sustained oscillations (van Ede et al., 2018). To study the transient events and the morphology of the waveforms (Mazaheri and Jensen, 2008; Cole and Voytek, 2017), which matter in cognition and for our understanding of pathologies (Jones, 2016; Cole et al., 2017), there is a clear need to go beyond traditionally employed signal processing methodologies (Cole and Voytek, 2018). For instance, a classic Fourier analysis fails to distinguish alpha-rhythms from mu-rhythms, which have the same peak frequency at around 10 Hz, but whose waveforms are different (Cole and Voytek, 2017; Hari and Puce, 2017).

The key to many modern statistical analyses of complex data such as natural images, sounds or neural time series is the estimation of data-driven representations. Dictionary learning is one family

of techniques, which consists in learning atoms (or patterns) that offer sparse data approximations. When working with long signals in which events can happen at any instant, one idea is to learn *shift-invariant* atoms. They can offer better signal approximations than generic bases such as Fourier or wavelets, since they are not limited to narrow frequency bands. Multiple approaches have been proposed to solve this shift-invariant dictionary learning problem, such as MoTIF (Jost et al., 2006), the sliding window matching (Gips et al., 2017), the adaptive waveform learning (Hitziger et al., 2017), or the learning of recurrent waveform (Brockmeier and Príncipe, 2016), yet they all have several limitations, as discussed in Jas et al. (2017). A more popular approach, especially in image processing, is the convolutional sparse coding (CSC) model (Jas et al., 2017; Pachitariu et al., 2013; Kavukcuoglu et al., 2010; Zeiler et al., 2010; Heide et al., 2015; Wohlberg, 2016b; Šorel and Šroubek, 2016; Grosse et al., 2007; Mailhé et al., 2008). The idea is to cast the problem as an optimization problem, representing the signal as a sum of convolutions between atoms and activation signals.

The CSC approach has been quite successful in several fields such as computer vision (Kavukcuoglu et al., 2010; Zeiler et al., 2010; Heide et al., 2015; Wohlberg, 2016b; Šorel and Šroubek, 2016), biomedical imaging (Jas et al., 2017; Pachitariu et al., 2013), and audio signal processing (Grosse et al., 2007; Mailhé et al., 2008), yet it was essentially developed for univariate signals. Interestingly, images can be multivariate such as color or hyper-spectral images, yet most CSC methods only consider gray scale images. To the best of our knowledge, the only reference to multivariate CSC is Wohlberg (2016a), where the author proposes two models well suited for 3-channel images. In the case of EEG and MEG recordings, neural activity is instantaneously and linearly spread across channels, due to Maxwell's equations (Hari and Puce, 2017). The same temporal patterns are reproduced on all channels with different intensities, which depend on each activity's location in the brain. To exploit this property, we propose to use a rank-1 constraint on each multivariate atom. This idea has been mentioned in (Barthélemy et al., 2012, 2013), but was considered less flexible than the full-rank model. Moreover, their proposed optimization techniques are not specific to shift-invariant models, and not scalable to long signals. Multivariate shift-invariant rank-1 decomposition of EEG has also been considered with matching pursuit (Durka et al., 2005), but without learning the atoms, which are fixed Gabor filters.

**Contribution** In this study, we develop a multivariate model for CSC, using a rank-1 constraint on the atoms to account for the instantaneous spreading of an electromagnetic source over all the channels. We also propose efficient optimization strategies, namely a locally greedy coordinate descent (LGCD, Moreau et al. 2018), and precomputation steps for faster gradient computations. We provide multiple numerical evaluations of our method, which show the highly competitive running time on both univariate and multivariate models, even when working with hundreds of channels. We also demonstrate the estimation performance of the multivariate model by recovering patterns on low signal-to-noise ratio (SNR) data. Finally, we illustrate our method with atoms learned on multivariate MEG data, that thanks to the rank-1 model can be localized in the brain for clinical or cognitive neuroscience studies.

**Notation** A multivariate signal with $T$ time points in $\mathbb{R}^P$ is noted $X \in \mathbb{R}^{P \times T}$, while $x \in \mathbb{R}^T$ is a univariate signal. We index time with brackets $X[t] \in \mathbb{R}^p$, while $X_i \in \mathbb{R}^T$ is the channel $i$ in $X$. For a vector $v \in \mathbb{R}^P$ we define the $\ell_q$ norm as $\|v\|_q = (\sum_i |v_i|^q)^{1/q}$, and for a multivariate signal $X \in \mathbb{R}^{P \times T}$, we define the time-wise $\ell_q$ norm as $\|X\|_q = (\sum_{t=1}^T \|X[t]\|_q^q)^{1/q}$. The transpose of a matrix $U$ is denoted by $U^\top$. For a multivariate signal $X \in \mathbb{R}^{P \times T}$, $X^{\reflectbox{7}}$ is obtained by reversal of the temporal dimension, *i.e.,* $X^{\reflectbox{7}}[t] = X[T + 1 - t]$. The convolution of two signals $z \in \mathbb{R}^{T-L+1}$ and $d \in \mathbb{R}^L$ is denoted by $z * d \in \mathbb{R}^T$. For $D \in \mathbb{R}^{P \times L}$, $z * D$ is obtained by convolving every row of $D$ by $z$. For $D' \in \mathbb{R}^{P \times L}$, $D \tilde{*} D' \in \mathbb{R}^{2L-1}$ is obtained by summing the convolution between each row of $D$ and $D'$: $D \tilde{*} D' = \sum_{p=1}^P D_p * D'_p$. We note $[a, b]$ the set of real numbers between $a$ and $b$, and $[\![a, b]\!]$ the set of integers between $a$ and $b$. We define $\widetilde{T}$ as $T - L + 1$.

## 2 Multivariate Convolutional Sparse Coding

In this section, we introduce the convolutional sparse coding (CSC) models used in this work. We focus on 1D-convolution, although these models can be naturally extended to higher order signals such as images by using the proper convolution operators.

**Univariate CSC**  The CSC formulation adopted in this work follows the shift-invariant sparse coding (SISC) model from Grosse et al. (2007). It is defined as follows:

$$\min_{\{d_k\}_k, \{z_k^n\}_{k,n}} \sum_{n=1}^{N} \frac{1}{2} \left\| x^n - \sum_{k=1}^{K} z_k^n * d_k \right\|_2^2 + \lambda \sum_{k=1}^{K} \|z_k^n\|_1 \,, \tag{1}$$

$$\text{s.t.} \quad \|d_k\|_2^2 \leq 1 \text{ and } z_k^n \geq 0 \,,$$

where $\{x^n\}_{n=1}^{N} \subset \mathbb{R}^T$ are $N$ observed signals, $\lambda > 0$ is the regularization parameter, $\{d_k\}_{k=1}^{K} \subset \mathbb{R}^L$ are the $K$ temporal atoms we aim to learn, and $\{z_k^n\}_{k=1}^{K} \subset \mathbb{R}^{\widetilde{T}}$ are $K$ signals of activations, a.k.a. the code associated with $x^n$. This model assumes that the coding signals $z_k^n$ are sparse, in the sense that only few entries are nonzero in each signal. In this work, we also assume that the entries of $z_k^n$ are positive, which means that the temporal patterns are present each time with the same polarity.

**Multivariate CSC**  The multivariate formulation uses an additional dimension on the signals and on the atoms, since the signal is recorded over $P$ channels (mapping to space locations):

$$\min_{\{D_k\}_k, \{z_k^n\}_{k,n}} \sum_{n=1}^{N} \frac{1}{2} \left\| X^n - \sum_{k=1}^{K} z_k^n * D_k \right\|_2^2 + \lambda \sum_{k=1}^{K} \|z_k^n\|_1, \tag{2}$$

$$\text{s.t.} \quad \|D_k\|_2^2 \leq 1 \text{ and } z_k^n \geq 0 \,,$$

where $\{X^n\}_{n=1}^{N} \subset \mathbb{R}^{P \times T}$ are $N$ observed multivariate signals, $\{D_k\}_{k=1}^{K} \subset \mathbb{R}^{P \times L}$ are the spatio-temporal atoms, and $\{z_k^n\}_{k=1}^{K} \subset \mathbb{R}^{\widetilde{T}}$ are the sparse activations associated with $X^n$.

**Multivariate CSC with rank-1 constraint**  This model is similar to the multivariate case but it adds a rank-1 constraint on the dictionary, $D_k = u_k v_k^\top \in \mathbb{R}^{P \times L}$, with $u_k \in \mathbb{R}^P$ being the pattern over channels and $v_k \in \mathbb{R}^L$ the pattern over time. The optimization problem boils down to:

$$\min_{\{u_k\}_k, \{v_k\}_k, \{z_k^n\}_{k,n}} \sum_{n=1}^{N} \frac{1}{2} \left\| X^n - \sum_{k=1}^{K} z_k^n * (u_k v_k^\top) \right\|_2^2 + \lambda \sum_{k=1}^{K} \|z_k^n\|_1 \,, \tag{3}$$

$$\text{s.t.} \quad \|u_k\|_2^2 \leq 1 \,, \|v_k\|_2^2 \leq 1 \text{ and } z_k^n \geq 0 \,.$$

The rank-1 constraint is consistent with Maxwell's equations and the physical model of electrophysiological signals like EEG or MEG, where each source is linearly spread instantaneously over channels with a constant topographic map (Hari and Puce, 2017). Using this assumption, one aims to improve the estimation of patterns under the presence of independent noise over channels. Moreover, it can help separating overlapped sources which are inherently rank-1 but whose sum is generally of higher rank. Finally, as explained below, several computations can be factorized to speed up computations.

**Noise model**  Note that our models use a Gaussian noise, whereas one can also use an alpha-stable noise distribution to better handle strong artifacts, as proposed by Jas et al. (2017). Importantly, our contribution is orthogonal to their work, and one can easily extend multivariate models to alpha-stable noise distributions, by using their EM algorithm and by updating the $\ell_2$ loss into a weighted $\ell_2$ loss in (3). Also, our experiments used artifact-free datasets, so the Gaussian noise model is appropriate.

## 3 Model estimation

Problems (1), (2) and (3) share the same structure. They are convex in each variable but not jointly convex. The resolution is done by using a block coordinate descent approach which minimizes alternately the objective function over one block of the variables. In this section, we describe this approach on the multivariate CSC with rank-1 constraint case (3), updating iteratively the activations $z_k^n$, the spatial patterns $u_k$, and the temporal pattern $v_k$.

### 3.1 $Z$-step: solving for the activations

Given $K$ *fixed* atoms $D_k$ and a regularization parameter $\lambda > 0$, the $Z$-step aims to retrieve the $NK$ activation signals $z_k^n \in \mathbb{R}^{\widetilde{T}}$ associated to the signals $X^n \in \mathbb{R}^{P \times T}$ by solving the following

---
**Algorithm 1:** Locally greedy coordinate descent (LGCD)
---
**Input:** Signal $X$, atoms $D_k$, number of segments $M$, stopping parameter $\epsilon > 0$, $z_k$ initialization
Initialize $\beta_k[t]$ with (5).
**repeat**
    **for** $m = 1$ **to** $M$ **do**
        Compute $z'_k[t] = \max\left(\frac{\beta_k[t]-\lambda}{\|D_k\|_2^2}, 0\right)$ for $(k,t) \in \mathcal{C}_m$
        Choose $(k_0, t_0) = \underset{(k,t)\in\mathcal{C}_m}{\arg\max} |z_k[t] - z'_k[t]|$
        Update $\beta$ with (6)
        Update the current point estimate $z_{k_0}[t_0] \leftarrow z'_{k_0}[t_0]$
**until** $\|z - z'\|_\infty < \epsilon$

---

$\ell_1$-regularized optimization problem:

$$\min_{\substack{\{z_k^n\}_{k,n} \\ z_k^n \geq 0}} \frac{1}{2} \left\| X^n - \sum_{k=1}^{K} z_k^n * D_k \right\|_2^2 + \lambda \sum_{k=1}^{K} \|z_k^n\|_1 \ . \tag{4}$$

This problem is convex in $z_k^n$ and can be efficiently solved. In Chalasani et al. (2013), the authors proposed an algorithm based on FISTA (Beck and Teboulle, 2009) to solve it. Bristow et al. (2013) introduced a method based on ADMM (Boyd et al., 2011) to compute efficiently the activation signals $z_k^n$. These two methods are detailed and compared by Wohlberg (2016b), which also made use of the fast Fourier transform (FFT) to accelerate the computations. Recently, Jas et al. (2017) proposed to use L-BFGS (Byrd et al., 1995) to improve on first order methods. Finally, Kavukcuoglu et al. (2010) adapted the greedy coordinate descent (GCD) to solve this convolutional sparse coding problem.

However, for long signals, these techniques can be quite slow due the computation of the gradient (FISTA, ADMM, L-BFGS) or the choice of the best coordinate to update in GCD, which are operations that scale linearly in $T$. A way to alleviate this limitation is to use a locally greedy coordinate descent (LGCD) strategy, presented recently in Moreau et al. (2018).

Note that problem (4) is independent for each signal $X^n$. The computation of each $z^n$ can thus be parallelized, independently of the technique selected to solve the optimization (Jas et al., 2017). Therefore, we omit the superscript $n$ in the following subsection to simplify the notation.

**Coordinate descent (CD)** The key idea of coordinate descent is to update our estimate of the solution one coordinate $z_k[t]$ at a time. For (4), it is possible to compute the optimal value $z'_k[t]$ of one coordinate $z_k[t]$ given that all the others are fixed. Indeed, the problem (4) restricted to one coordinate has a closed-form solution given by:

$$z'_k[t] = \max\left(\frac{\beta_k[t]-\lambda}{\|D_k\|_2^2}, 0\right), \text{ with } \beta_k[t] = \left[D_k^{\natural} \tilde{*} \left(X - \sum_{l=1}^{K} z_l * D_l + z_k[t]e_t * D_k\right)\right][t] \tag{5}$$

where $e_t \in \mathbb{R}^{\widetilde{T}}$ is the canonical basis vector with value 1 at index $t$ and 0 elsewhere. When updating the coefficient $z_{k_0}[t_0]$ to the value $z'_{k_0}[t_0]$, $\beta$ is updated with:

$$\beta_k^{(q+1)}[t] = \beta_k^{(q)}[t] + (D_{k_0}^{\natural} \tilde{*} D_k)[t - t_0](z_{k_0}[t_0] - z'_{k_0}[t_0]), \qquad \forall(k,t) \neq (k_0, t_0) \ . \tag{6}$$

The term $(D_{k_0}^{\natural} \tilde{*} D_k)[t - t_0]$ is zero for $|t - t_0| \geq L$. Thus, only $K(2L-1)$ coefficients of $\beta$ need to be changed (Kavukcuoglu et al., 2010). The CD algorithm updates at each iteration a coordinate to this optimal value. The coordinate to update can be chosen with different strategies, such as the cyclic strategy which iterates over all coordinates (Friedman et al., 2007), the randomized CD (Nesterov, 2010; Richtárik and Takáč, 2014) which chooses a coordinate at random for each iteration, or the greedy CD (Osher and Li, 2009) which chooses the coordinate the farthest from its optimal value.

**Locally greedy coordinate descent (LGCD)** The choice of a coordinate selection strategy results of a tradeoff between the computational cost of each iteration and the improvement it provides. For cyclic and randomized strategies, the iteration complexity is $\mathcal{O}(KL)$ as the coordinate selection can

Table 1: Computational complexities of each step

| Step | Computation | Computed | Rank-1 | Full-rank |
|------|-------------|----------|--------|-----------|
| $Z$-step | $\beta$ initialization | once | $NKT(L+P)$ | $NKT(LP)$ |
| $Z$-step | Precomputation | once | $K^2L(L+P)$ | $K^2L(LP)$ |
| $Z$-step | M coordinate updates | multiple times | $MKL$ | $MKL$ |
| $D$-step | $\Phi$ precomputation | once | $NKTLP$ | $NKTLP$ |
| $D$-step | $\Psi$ precomputation | once | $NK^2TL$ | $NKTLP$ |
| $D$-step | Gradient evaluation | multiple times | $K^2L(L+P)$ | $K^2L(LP)$ |
| $D$-step | Function evaluation | multiple times | $K^2L(L+P)$ | $K^2L(LP)$ |

be performed in constant time. The greedy selection of a coordinate is more expensive as it is linear in the signal length $\mathcal{O}(K\widetilde{T})$. However, greedy selection is more efficient iteration-wise (Nutini et al., 2015). Moreau et al. (2018) proposed to consider a locally greedy selection strategy for CD. The coordinate to update is chosen greedily in one of $M$ subsegments of the signal, *i.e.,* at iteration $q$, the selected coordinate is:

$$(k_0, t_0) = \underset{(k,t) \in \mathcal{C}_m}{\arg\max} |z_k[t] - z'_k[t]|, \qquad m \equiv q \,(\text{mod } M) + 1 , \qquad (7)$$

with $\mathcal{C}_m = [\![1, K]\!] \times [\![(m-1)\widetilde{T}/M, m\widetilde{T}/M]\!]$. With this strategy, the coordinate selection complexity is linear in the length of the considered subsegment $\mathcal{O}(K\widetilde{T}/M)$. By choosing $M = \lfloor \widetilde{T}/(2L-1) \rfloor$, the complexity of update is the same as the complexity of random and cyclic coordinate selection, $\mathcal{O}(KL)$. We detail the steps of LGCD in Algorithm 1. This algorithm is particularly efficient when the $z_k$ are sparser. Indeed, in this case, only few coefficients need to be updated in the signal, resulting in a low number of iterations. Computational complexities are detailed in Table 1.

**Relation with matching pursuit (MP)** Note that the greedy CD is strongly related to the well-known matching pursuit (MP) algorithm (Locatello et al., 2018). The main difference is that MP solves a slightly different problem, where the $\ell_1$ regularization is replaced with an $\ell_0$ constraint. Therefore, the size of the support is a fixed parameter in MP, whereas it is controlled by the regularization parameter $\lambda$ in our case. In term of algorithm, both methods update one coordinate at a time selected greedily, but MP does not apply a soft-thresholding in (5).

## 3.2 $D$-step: solving for the atoms

Given $KN$ *fixed* activation signals $z_k^n \in \mathbb{R}^{\widetilde{T}}$, associated to signals $X^n \in \mathbb{R}^{P \times T}$, the $D$-step aims to update the $K$ spatial patterns $u_k \in \mathbb{R}^P$ and $K$ temporal patterns $v_k \in \mathbb{R}^L$, by solving:

$$\min_{\substack{\|u_k\|_2 \leq 1 \\ \|v_k\|_2 \leq 1}} E, \qquad \text{where} \qquad E \triangleq \sum_{n=1}^{N} \frac{1}{2} \|X^n - \sum_{k=1}^{K} z_k^n * (u_k v_k^\top)\|_2^2 \quad . \qquad (8)$$

The problem (8) is convex in each block of variables $\{u_k\}_k$ and $\{v_k\}_k$, but not jointly convex. Therefore, we optimize first $\{u_k\}_k$, then $\{v_k\}_k$, using in both cases a projected gradient descent with an Armijo backtracking line-search (Wright and Nocedal, 1999) to find a good step size. These steps are detailed in Algorithm A.1.

**Gradient relative to $u_k$ and $v_k$** The gradient of $E$ relatively to $\{u_k\}_k$ and $\{v_k\}_k$ can be computed using the chain rule. First, we compute the gradient relatively to a full atom $D_k = u_k v_k^\top \in \mathbb{R}^{P \times L}$:

$$\nabla_{D_k} E = \sum_{n=1}^{N} (z_k^n)^{\backprime} * \left( X^n - \sum_{l=1}^{K} z_l^n * D_l \right) = \Phi_k - \sum_{l=1}^{K} \Psi_{k,l} * D_l , \qquad (9)$$

where we reordered this expression to define $\Phi_k \in \mathbb{R}^{P \times L}$ and $\Psi_{k,l} \in \mathbb{R}^{2L-1}$. These terms are both constant during a $D$-step and can thus be precomputed to accelerate the computation of the

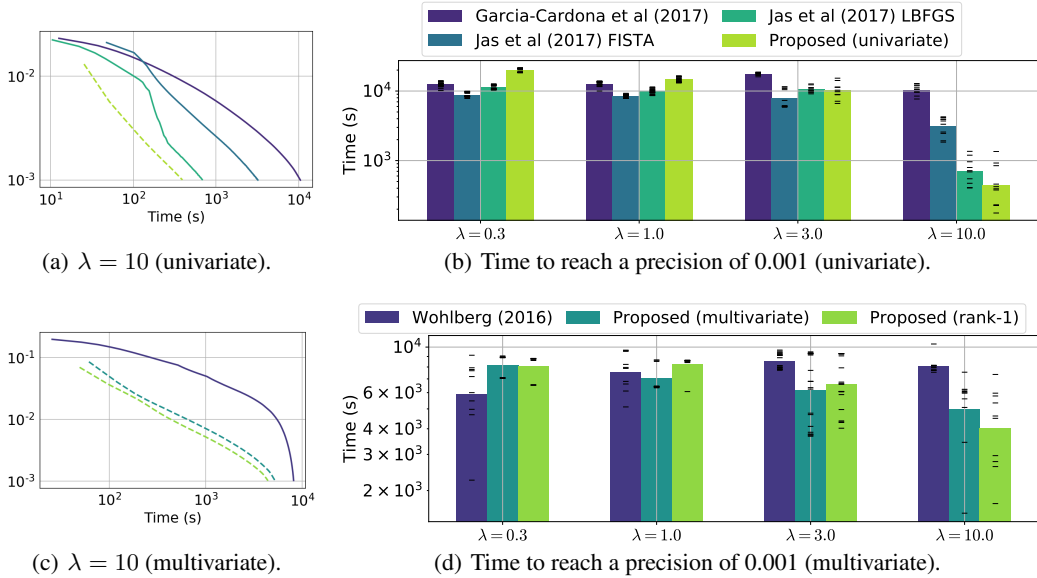

Figure 1: Comparison of state-of-the-art univariate (a, b) and multivariate (c, d) methods with our approach. (a) Convergence plot with the objective function relative to the obtained minimum, as a function of computational time. (b) Time taken to reach a relative precision of $10^{-3}$, for different regularization parameters $\lambda$. (c, d) Same as (a, b) in the multivariate setting $P = 5$.

gradients and the cost function $E$. We detail these computations in the supplementary materials (see Section A.1). Computational complexities are detailed in Table 1. Note that the dependence in $T$ is present *only* in the precomputations, which makes the following iterations very fast. Without precomputations, the complexity of *each* gradient computation in the $D$-step would be $\mathcal{O}(NKTLP)$.

### 3.3 Initialization

The activations sub-problem ($Z$-step) is regularized with a $\ell_1$-norm, which induces sparsity: the higher the regularization parameter $\lambda$, the higher the sparsity. Therefore, there exists a value $\lambda_{max}$ above which the sub-problem solution is always zeros (Hastie et al., 2015). As $\lambda_{max}$ depends on the atoms $D_k$ and on the signals $X^n$, its value changes after each $D$-step. In particular, its value might change a lot between the initialization and the first $D$-step. This is problematic since we cannot use a regularization $\lambda$ above this initial $\lambda_{max}$, even though the following $\lambda_{max}$ might be higher.

The standard strategy to initialize CSC methods is to generate random atoms with Gaussian white noise. However, as these atoms generally poorly correlate with the signals, the initial value of $\lambda_{max}$ is low compared to the following ones. For example, on the MEG dataset described later on, we found that the initial $\lambda_{max}$ is about $1/3$ of the following ones in the univariate case, with $L = 32$. On the multivariate case, it is even more problematic as with $P = 204$, we could have an initial $\lambda_{max}$ as low as $1/20$ of the following ones.

To fix this problem, we propose to initialize the dictionary with random chunks of the signal, projecting each chunk on a rank-1 approximation using singular value decomposition. We noticed on the MEG dataset that the initial $\lambda_{max}$ was then about the same value as the following ones, which enables the use of higher regularization parameters. We used this scheme in all our experiments.

## 4 Experiments

All numerical experiments were run using Python (Python Software Foundation, 2017) and our code is publicly available online at `https://alphacsc.github.io/`.

**Speed performance** To illustrate the performance of our optimization strategy, we monitored its convergence speed on a real MEG dataset. The somatosensory dataset from the MNE software (Gram-

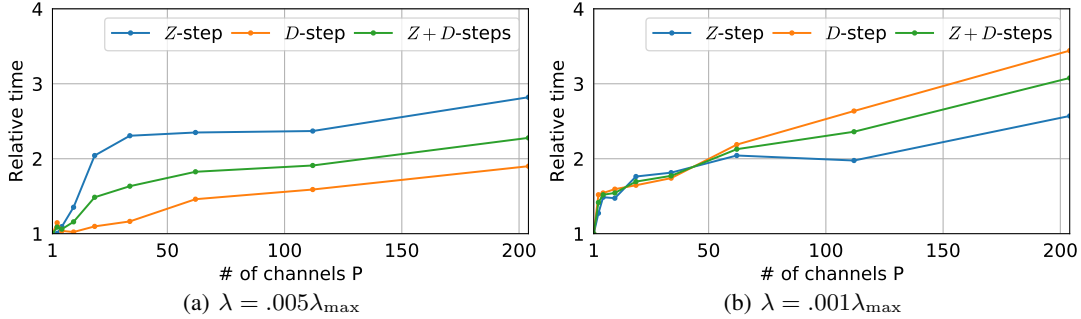

(a) $\lambda = .005\lambda_{\max}$   (b) $\lambda = .001\lambda_{\max}$

Figure 2: Timings of $Z$ and $D$ updates when varying the number of channels $P$. The scaling is sublinear with $P$, due to the precomputation steps in the optimization.

fort et al., 2013, 2014) contains responses to median nerve stimulation. We consider only gradiometers channels and we used the following parameters: $T = 134\,700$, $N = 2$, $K = 8$, and $L = 128$.

First we compared our strategy against three state-of-the-art *univariate* CSC solvers available online. The first was developed by Garcia-Cardona and Wohlberg (2017) and is based on ADMM. The second and third were developed by Jas et al. (2017), and are respectively based on FISTA and L-BFGS. All solvers shared the same objective function, but as the problem is non-convex, the solvers are not guaranteed to reach the same local minima, even though we started from the same initial settings. Hence, for a fair comparison, we computed the convergence curves relative to each local minimum, and averaged them over 10 different initializations. The results, presented in Figure 1(a, b), demonstrate the competitiveness of our method, for reasonable choices of $\lambda$. Indeed, a higher regularization parameter leads to sparser activations $z_k^n$, on which LGCD is particularly efficient.

Then, we also compared our method against a multivariate ADMM solver developed by Wohlberg (2016a). As this solver was quite slow on these long signals, we limited our experiments to $P = 5$ channels. The results, presented in Figure 1(c, d), show that our method is faster than the competing method for large $\lambda$. More benchmarks are available in the supplementary materials.

**Scaling with the number of channels**   The multivariate model involves an extra dimension $P$ but its impact on the computational complexity of our solver is limited. Figure 2 shows the average running times of the $Z$-step and the $D$-step. Timings are normalized *w.r.t.* the timings for a single channel. The running times are computed using the same signals from the somatosensory dataset, with the following parameters: $T = 26\,940$, $N = 10$, $K = 2$, $L = 128$. We can see that the scaling of these three operations is sub-linear in $P$. For the $Z$-step, only the initial computations for the first $\beta_k$ and the constants $D_k^\uparrow \tilde{*} D_l$ depend linearly on $P$ so that the complexity increase is limited compared to the complexity of solving the optimization problem (4). For the $D$-step, the scaling to compute the gradients is linear with $P$. However, the most expensive operations here are the computation of the constant $\Psi_k$, which does not on $P$ .

**Finding patterns in low SNR signals**   Since the multivariate model has access to more data, we would expect it to perform better compared to the univariate model especially for low SNR signals. To demonstrate this, we compare the two models when varying the number of channels $P$ and the SNR of the data. The original dictionary contains two temporal patterns, a square and a triangle, presented in Figure 3(a). The spatial maps are designed with a sine and a cosine, and the first channel's amplitude is forced to 1 to make sure both atoms are present even with only one channel. The signals are obtained by convolving the atoms with activation signals $z_k^n$, where the activation locations are sampled uniformly in $[\![1, \widetilde{T}]\!] \times [\![1, K]\!]$ with 5% non-zero activations, and the amplitudes are uniformly sampled in $[0, 1]$. Then, a Gaussian white noise with variance $\sigma$ is added to the signal. We fixed $N = 100$, $L = 64$ and $\widetilde{T} = 640$ for our simulated signals. We can see in Figure 3(a) the temporal patterns recovered for $\sigma = 10^{-3}$ using only one channel and using 5 channels. While the patterns recovered with one channel are very noisy, the multivariate model with rank-1 constraint recovers the original atoms accurately. This can be expected as the univariate model is ill-defined in this situation, where some atoms are superimposed. For the rank-1 model, as the atoms have different spatial maps, the problem is easier. Then, we evaluate the learned temporal atoms. Due to permutation and sign ambiguity, we compute the $\ell_2$-norm of the difference between the temporal

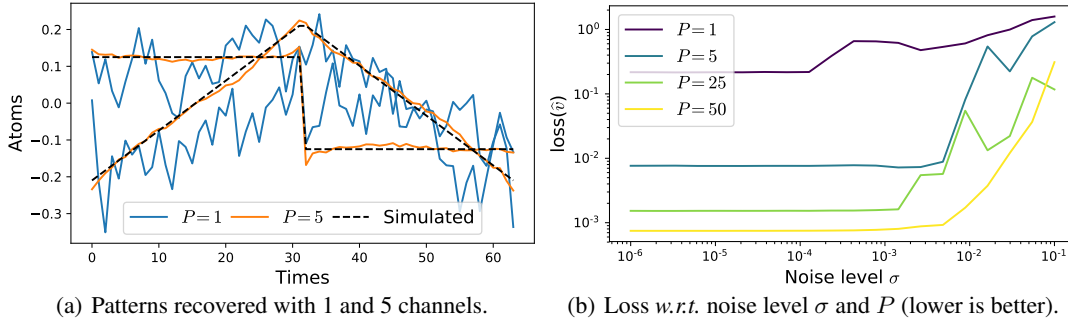

(a) Patterns recovered with 1 and 5 channels.

(b) Loss *w.r.t.* noise level $\sigma$ and $P$ (lower is better).

Figure 3: (*a*) Patterns recovered with $P = 1$ and $P = 5$. The signals were generated with the two simulated temporal patterns and with $\sigma = 10^{-3}$. (*b*) Evolution of the recovery loss with $\sigma$ for different values of $P$. Using more channels improves the recovery of the original patterns.

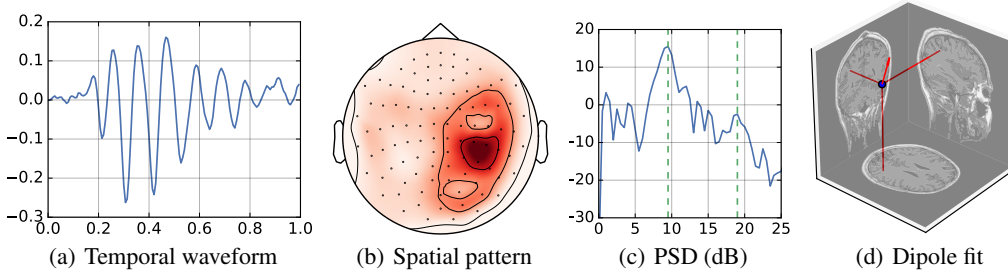

(a) Temporal waveform     (b) Spatial pattern     (c) PSD (dB)     (d) Dipole fit

Figure 4: Atom revealed using the MNE somatosensory data. Note the non-sinusoidal comb shape of the mu rhythm. This atom has been manually selected, and other atoms are presented in Figure B.4.

pattern $\widehat{v}_k$ and the ground truths, $v_k$ or $-v_k$, for all permutations $\mathfrak{S}(K)$ *i.e.*,

$$\text{loss}(\widehat{v}) = \min_{s \in \mathfrak{S}(K)} \sum_{k=1}^{K} \min \left( \|\widehat{v}_{s(k)} - v_k\|_2^2, \|\widehat{v}_{s(k)} + v_k\|_2^2 \right) \quad . \tag{10}$$

Multiple values of $\lambda$ were tested and the best loss is reported in Figure 3(b) for varying noise levels $\sigma$. We observe that independently of the noise level, the multivariate rank-1 model outperforms the univariate one. This is true even for good SNR, as using multiple channels disambiguates the separation of overlapped patterns.

**Examples of atoms in real MEG signals:**    We show the results of our algorithm on experimental data, using the MNE somatosensory dataset (Gramfort et al., 2013, 2014). This dataset contains MEG recordings of one patient receiving median nerve stimulations. Here we first extract $N = 103$ trials from the data. Each trial lasts 6 s with a sampling frequency of 150 Hz ($T = 900$). We selected only gradiometer channels, leading to $P = 204$ channels. The signals were notch-filtered to remove the power-line noise, and high-pass filtered at 2 Hz to remove the low-frequency trend, *i.e.* to remove low frequency drift artifacts which contribute a lot to the variance of the raw signals. We learned $K = 40$ atoms with $L = 150$ using a rank-1 multivariate CSC model, with a regularization $\lambda = 0.2\lambda_{max}$.

Figure 4(a) shows a recovered non-sinusoidal brain rhythm which resembles the well-known mu-rhythm. The mu-rhythm has been implicated in motor-related activity (Hari, 2006) and is centered around 9–11 Hz. Indeed, while the power is concentrated in the same frequency band as the alpha, it has a very different spatial topography (Figure 4(b)). In Figure 4(c), the power spectral density (PSD) shows two components of the mu-rhythm – one at around 9 Hz, and a harmonic at 18 Hz as previously reported in (Hari, 2006). Based on our analysis, it is clear that the 18 Hz component is simply a harmonic of the mu-rhythm even though a Fourier-based analysis could lead us to falsely conclude that the data contained beta-rhythms. Finally, due to the rank-1 nature of our atoms, it is straightforward to fit an equivalent current dipole (Tuomisto et al., 1983) to interpret the origin of the signal. Figure 4(d) shows that the atom does indeed localize in the primary somatosensory cortex, or the so-called S1 region with a 59.3% goodness of fit. For results on more MEG datasets, see Section B.2. It notably includes mu-shaped atoms from S2.

# 5 Conclusion

Many neuroscientific debates today are centered around the morphology of the signals under consideration. For instance, are alpha-rhythms asymmetric (Mazaheri and Jensen, 2008) ? Are frequency specific patterns the result of sustained oscillations or transient bursts (van Ede et al., 2018) ? In this paper, we presented a multivariate extension to the CSC problem applied to MEG data to help answer such questions. In the original CSC formulation, the signal is expressed as a convolution of atoms and their activations. Our method extends this to the case of multiple channels and imposes a rank-1 constraint on the atoms to account for the instantaneous propagation of electromagnetic fields. We demonstrate the usefulness of our method on publicly available multivariate MEG data. Not only are we able to recover neurologically plausible atoms, but also we are able to find temporal waveforms which are non-sinusoidal. Empirical evaluations show that our solvers are significantly faster compared to existing CSC methods even for the univariate case (single channel). The algorithm scales sublinearly with the number of channels which means it can be employed even for dense sensor arrays with 200-300 sensors, leading to better estimation of the patterns and their origin in the brain.

## Acknowledgment

This work was supported by the ERC Starting Grant SLAB ERC-YStG-676943 and by the ANR THALAMEEG ANR-14-NEUC-0002-01.

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
