[Supplementary Material]

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

# A  Optimization details

In this section, we give more details about the optimization procedures used to speed-up both $D$-step and $Z$-step.

## A.1  Details on the $D$-step optimization

First, let's recall the objective function, as introduced in Section 3.2:

$$E \triangleq \sum_{n=1}^{N} \frac{1}{2} \|X^n - \sum_{k=1}^{K} z_k^n * (u_k v_k^\top)\|_2^2, \tag{A.1}$$

which we optimize under the constraints $\|u_k\|_2^2 \leq 1$ and $\|v_k\|_2^2 \leq 1$.

To compute the gradient of $E$ relatively to a full atom $D_k = u_k v_k^\top \in \mathbb{R}^{P \times L}$, we introduce some constants $\Phi_k$ and $\Psi_{k,l}$, which are constant during the entire $D$-step:

$$\nabla_{D_k} E = \sum_{n=1}^{N} (z_k^n)^\text{¶} * \left(X^n - \sum_{l=1}^{K} z_l^n * D_l\right) = \Phi_k - \sum_{l=1}^{K} \Psi_{k,l} * D_l \tag{A.2}$$

Indeed, we have:

$$\nabla_{D_k} E[t] = \sum_{n=1}^{N} \left((z_k^n)^\text{¶} * \left(X^n - \sum_{l=1}^{K} z_l^n * D_l\right)\right)[t] \tag{A.3}$$

$$= \sum_{n=1}^{N} \sum_{\tau=1}^{\widetilde{T}} z_k^n[\tau] \left(X^n - \sum_{l=1}^{K} z_l^n * D_l\right)[t + \tau - 1] \tag{A.4}$$

$$= \sum_{n=1}^{N} \sum_{\tau=1}^{\widetilde{T}} z_k^n[\tau] \left(X^n[t + \tau - 1] - \sum_{l=1}^{K} \sum_{\tau'=1}^{L} z_l^n[\tau'] D_l[t + \tau - \tau']\right) \tag{A.5}$$

$$= \Phi_k[t] - \sum_{l=1}^{K} \sum_{\tau'=1}^{L} \left(\sum_{n=1}^{N} \sum_{\tau=1}^{\widetilde{T}} z_k^n[\tau] z_l^n[t + \tau - \tau']\right) D_l[\tau'] \tag{A.6}$$

$$= \Phi_k[t] - \sum_{l=1}^{K} \sum_{\tau'=1}^{L} \Psi_{k,l}[t + 1 - \tau'] D_l[\tau'] \tag{A.7}$$

$$= \Phi_k[t] - \sum_{l=1}^{K} (\Psi_{k,l} * D_l)[t] \tag{A.8}$$

where $\Phi_k \in \mathbb{R}^{P \times L}$ are computed with:

$$\Phi_k[t] = \sum_{n=1}^{N} \sum_{\tau=1}^{\widetilde{T}} z_k^n[\tau] X^n[t + \tau - 1], \quad \forall t \in [\![1, L]\!], \tag{A.9}$$

and where $\Psi_{k,l} \in \mathbb{R}^{2L-1}$ are computed with:

$$\Psi_{k,l}[t] = \sum_{n=1}^{N} \sum_{\tau=1}^{\widetilde{T}} z_k^n[\tau] z_l^n[t + \tau - 1], \quad \forall t \in [\![1, 2L - 1]\!]. \tag{A.10}$$

Note that in the last equation (A.10), the sum only concerns the defined terms, *i.e.*, $(t + \tau - 1) \in [\![1, \widetilde{T}]\!]$. The computational complexities of $\Phi_k$ and $\Psi_{k,l}$ are respectively $\mathcal{O}(NLTKP)$ and $\mathcal{O}(NLTK^2)$.

Then, the gradients relative to $u_k$ and $v_k$ are obtained using the chain rule,

$$\nabla_{u_k} E = (\nabla_{D_k} E) v_k \quad \in \mathbb{R}^P, \tag{A.11}$$

$$\nabla_{v_k} E = u_k^\top (\nabla_{D_k} E) \quad \in \mathbb{R}^L, \tag{A.12}$$

**Algorithm A.1:** Projected gradient descent for updating $\{u_k\}$ and $\{v_k\}$.

---

**Input :** Signals $\{X^n\}_n$, activations $\{z_k^n\}_{k,n}$, stopping parameter $\epsilon > 0$,
　　　　initial estimate $\{u_k\}_k$ and $\{v_k\}_k$
Initialize $\Phi_k$ with (A.9) and $\Psi_{k,l}$ with (A.10), for $k, l \subset [\![1, K]\!]$.
**repeat**
　　Compute with (A.11) for $k \in [\![1, K]\!]$, $G_k = \nabla_{u_k} E$,
　　Update the estimate with $\{u_k^{(q+1)}\}_k \leftarrow$ **to** Armijo($\{u_k^{(q)}\}_k, G_k, E$)
**until** $\sum_{k=1}^{K} \left\| u_k^{(q+1)} - u_k^{(q)} \right\|_1 < \epsilon$
Set $\{u_k\}_k \leftarrow \{u_k^{(q)}\}_k$
**repeat**
　　Compute with (A.12) for $k \in [\![1, K]\!]$, $G_k = \nabla_{v_k} E$,
　　Update the estimate with $\{v_k^{(q+1)}\}_k \leftarrow$ **to** Armijo($\{v_k^{(q)}\}_k, G_k, E$)
**until** $\sum_{k=1}^{K} \left\| v_k^{(q+1)} - v_k^{(q)} \right\|_1 < \epsilon$
Set $\{v_k\}_k \leftarrow \{v_k^{(q)}\}_k$
**return** $\{u_k\}_k$ and $\{v_k\}_k$

---

and $E$ can be computed, up to a constant term $C$ , with the following

$$E = \sum_{k=1}^{K} u_k^\top (\nabla_{D_k} E) v_k + C \ . \tag{A.13}$$

Algorithm A.1 details the different step used in our algorithm to update $\{u_k\}_k$ and $\{v_k\}_k$.

## A.2  Details on the $Z$-step optimization

### A.2.1  The coordinate update

**Proposition 1.** *The optimal update $z'_{k_0}[t_0]$ of the coefficient $(k_0, t_0)$ is given by*

$$z'_{k_0}[t_0] = \frac{1}{\|D_{k_0}\|_2^2} \max \left( \beta_{k_0}[t_0] - \lambda, 0 \right) \ ,$$

*with $\beta_{k_0}[t_0] = D_{k_0}^{\backprime} \tilde{\ast} \left( X - \sum_{k=1}^{K} z_k \ast D_k + z_{k_0}[t_0] e_{t_0} \ast D_{k_0} \right) [t_0]$ and where $e_{t_0}$ is the canonical vector in $\mathbb{R}^{\widetilde{T}}$ with value 1 in $t_0$ and value 0 elsewhere.*

*Proof.* For $y \in \mathbb{R}^+$, we will denote $e_{k_0, t_0}(y)$ the cost difference between our current solution estimate $z_k$ and the signal $z_k^{(1)}$ where the coefficient $z_{k_0}[t_0]$ has been replaced by $y$, *i.e.*,

$$z_k^{(1)}[t] = \begin{cases} y, & \text{if } (k, t) = (k_0, t_0) \\ z_k[t], & \text{elsewhere} \end{cases} \ .$$

Let $\alpha_{k_0}[t] = (X - \sum_{k=1}^{K} z_k \ast D_k)[t] + D_{k_0}[t - t_0] z_{k_0}[t_0]$ for all $t \in [\![0, T-1]\!]$. This quantity denotes the residual when $z_{k_0}[t_0]$ is set to 0. It is important to note that it can be re-written as,

$$\alpha_k[t] = \left( X - \sum_{k=1}^{K} z_k \ast D_k + z_{k_0}[t_0] e_{t_0} \ast D_{k_0} \right) [t]$$

and thus, $\beta_{k_0}[t_0] = \left(D_{k_0}^{\natural} \tilde{*} \alpha_{k_0}\right)[t_0]$. The cost difference $e_{k_0,t_0}(y)$ is,

$$e_{k_0,t_0}(y) = \frac{1}{2}\sum_{t=0}^{T-1}\left(X - \sum_{k=1}^{K} z_k * D_k\right)^2[t] + \lambda\sum_{k=1}^{K}\|z_k\|_1 - \frac{1}{2}\sum_{t=0}^{T-1}\left(X - \sum_{k=1}^{K} z_k^{(1)} * D_k\right)^2[t] + \lambda\sum_{k=1}^{K}\|z_k^{(1)}\|_1$$

$$= \frac{1}{2}\sum_{t=0}^{T-1}\left(\alpha_{k_0}[t] - D_{k_0}[t-t_0]z_{k_0}[t_0]\right)^2 - \frac{1}{2}\sum_{t=0}^{T-1}\left(\alpha_{k_0}[t] - D_{k_0}[t-t_0]y\right)^2 + \lambda(|z_{k_0}[t_0]| - |y|)$$

$$= \frac{1}{2}\sum_{t=0}^{T-1} D_{k_0}[t-t_0]^2(z_{k_0}[t_0]^2 - y^2) - \sum_{t=0}^{T-1}\alpha_{k_0}[t]D_{k_0}[t-t_0](z_{k_0}[t_0] - y) + \lambda(|z_{k_0}[t_0]| - |y|)$$

$$= \frac{\|D_{k_0}\|_2^2}{2}(z_{k_0}[t_0]^2 - y^2) - \underbrace{(D_{k_0}^{\natural} \tilde{*} \alpha_{k_0})[t_0]}_{\beta_{k_0}[t_0]}(z_{k_0}[t_0] - y) + \lambda(|z_{k_0}[t_0]| - |y|)$$

Using this result, we can derive the optimal value $z'_{k_0}[t_0]$ to update the coefficient $(k_0, t_0)$ as the solution of the following optimization problem:

$$z'_{k_0}[t_0] = \arg\max_{y\in\mathbb{R}^+} e_{k_0,t_0}(y) \sim \arg\min_{u\in\mathbb{R}^+} \frac{\|D_{k_0}\|_2^2}{2}\left(y - \frac{\beta_{k_0}[t_0]}{\|D_{k_0}\|_2^2}\right)^2 + \lambda y . \qquad \text{(A.14)}$$

Simple computations show the desired result, *i.e.*,

$$z'_{k_0}[t_0] = \frac{1}{\|D_{k_0}\|_2^2}\max(\beta_{k_0}[t_0] - \lambda, 0)$$

.                                                                                                                    □

### A.2.2 The $\beta$ update

**Proposition 2.** *When updating the coefficient $z_{k_0}[t_0]$ to the value $z'_{k_0}[t_0]$, $\beta$ is updated with:*

$$\beta_k^{(q+1)}[t] = \beta_k^{(q)}[t] + (D_{k_0}^{\natural} \tilde{*} D_k)[t - t_0](z_{k_0}[t_0] - z'_{k_0}[t_0]), \qquad \forall(k,t) \neq (k_0, t_0) . \qquad \text{(A.15)}$$

*Proof.* The value of $\beta_{k_0}[t_0]$ is independent of the value of $z_{k_0}[t_0]$. Indeed, the term $z_{k_0}[t_0]e_{t_0} * D_{k_0}$ cancel the contribution of $z_{k_0}[t_0]$ in the convolution $z_{k_0} * D_{k_0}$. Thus, when updating the value of the coefficient $z_{k_0}[t_0]$, $\beta_{k_0}[t_0]$ is not updated.

We denote $z_k^{(q+1)}$ the activation signal where the coefficient $z_{k_0}[t_0]$ as been updated to $z'_{k_0}[t_0]$, *i.e.*, ,

$$z_k^{(q+1)}[t] = \begin{cases} z'_{k_0}[t_0], & \text{if } (k,t) = (k_0, t_0) \\ z_k[t], & \text{elsewhere} \end{cases} .$$

For $(k,t) \neq (k_0, t_0)$,

$$\beta_k^{(q+1)}[t] = \left[D_k^{\natural} \tilde{*} \left(X - \sum_{l=1}^{K} z_l^{(1)} * D_l + z_k[t]e_t * D_k\right)\right][t]$$

$$= \left[D_k^{\natural} \tilde{*} \left(X - \sum_{l=1}^{K} z_l * D_l + z_k[t]e_t * D_k + (z_{k_0}[t_0] - z'_{k_0}[t_0])e_{t_0} * D_k\right)\right][t]$$

$$= \left[D_k^{\natural} \tilde{*} \left(X - \sum_{l=1}^{K} z_l * D_l + z_k[t]e_t * D_k\right)\right][t] + \left[D_k^{\natural} \tilde{*} \left((z_{k_0}[t_0] - z'_{k_0}[t_0])e_{t_0} * D_k\right)\right][t]$$

$$= \beta_k^{(q)}[t] + (z_{k_0}[t_0] - z'_{k_0}[t_0])\left[D_k^{\natural} \tilde{*} (e_{t_0} * D_k)\right][t]$$

$$= \beta_k^{(q)}[t] + (D_k^{\natural} \tilde{*} D_k)[t - t_0](z_{k_0}[t_0] - z'_{k_0}[t_0])$$

With this relation, it is possible to keep $\beta_k$ up to date with few operation after each coordinate update.                                                                            □

(a) Shorter atoms, univariate.

(b) Shorter atoms, multivariate ($P = 5$).

Figure B.1: Comparison of state-of-the-art methods with our approach. Here we used shorter ($L = 16$ instead of $L = 128$) atoms.

### A.2.3 Precomputation for $D_k^{\talloblong} \tilde{\ast} D_l$

Similarly to the $D$-step precomputations, we can precompute $D_k^{\talloblong} \tilde{\ast} D_l \in \mathbb{R}^{2L-1}$ to speed up the LGCD iterations during the $Z$-step. We have:

$$(D_k^{\talloblong} \tilde{\ast} D_l)[t] = \sum_{p=1}^{P} \sum_{\tau=1}^{L} D_{k,p}[\tau] D_{l,p}[t+\tau-1], \quad \forall t \in [\![1, 2L-1]\!]. \tag{A.16}$$

In the case of the rank-1 constraint model, we can factorize the computation with:

$$(D_k^{\talloblong} \tilde{\ast} D_l)[t] = \left( \sum_{p=1}^{P} u_{k,p} u_{l,p} \right) \sum_{\tau=1}^{L} v_k[\tau] v_l[t+\tau-1], \quad \forall t \in [\![1, 2L-1]\!]. \tag{A.17}$$

The computational complexities are respectively $\mathcal{O}\left(K^2 L^2 P\right)$ and $\mathcal{O}\left(K^2 L(L+P)\right)$.

## B  Additional Experiments

### B.1  Speed performance

We present here more benchmarks as described in section 4, yet with different settings.

First we used shorter atoms of length $L = 16$ instead of $L = 128$, and results are presented in Figure B.1. They confirm the competitiveness of our method, especially when using large regularization parameters. On these problems, the maximum possible regularization $\lambda_{max}$ was around 90.

Then, we used shorter signals of length $T = 13\,470$ instead of $T = 134\,700$ and results are presented in Figure B.2. They also confirm the competitiveness of our method, except with small regularization parameters. However, as the maximum possible regularization $\lambda_{max}$ was around 90, we question the practical use of these low values, which would poorly enforce the sparsity constraint.

### B.2  Somatosensory dataset

In Figure 4(d), we showed mu-shaped atoms in the *primary* somatosensory region for the MNE somatosensory dataset. Intriguingly, we also find such atoms in the *secondary* somatosensory region, also known as S2. One such atom is shown in Figure B.3.

(a) Shorter signals, univariate.

(b) Shorter signals, multivariate ($P = 5$).

Figure B.2: Comparison of state-of-the-art methods with our approach. Here we used shorter signals ($T = 13\,470$ instead of $T = 134\,700$).

Figure B.3: Atom in the S2 region revealed in the MNE somatosensory data. A. The temporal waveform, and its corresponding B. Spatial pattern, C. The Power Spectral Density (PSD), and D. the dipole fit in the S2 region.

## B.3    Sample dataset

In addition to the MNE somatosensory dataset, we also analyzed the MNE sample dataset Gramfort et al. (2013, 2014). In this case, we used $N = 1$, and the number of time points $T = 41584$ corresponds to 278 s of recording sampled at 150.15 Hz. The magnetometer channels are selected so that the number of channels $P = 102$. We learn $K = 25$ atoms. The sample data is lowpass filtered at 40 Hz, and highpass filtered at 1 Hz.

In Figure B.6, we show the atoms learned on the MNE sample data. Figure B.6.A shows the temporal waveforms of these atoms and Figure B.6.C shows the corresponding spatial pattern for a selection of the total atoms. As expected, we are able to recover latent components corresponding to ocular (3rd row) and cardiac artifacts (4th row). Indeed, the ocular artifacts displays the prototypical dipolar pattern in the frontal channels. In Figure B.6.B, we also show the sparse activations associated with the atoms.

More interestingly, we also recover an oscillatory waveform (first row) which appears to originate due to a dipole below the parietal channels at around a frequency of 30 Hz. We confirm this in Figure B.7 using a dipole fit. Indeed, the atom does originate in the parietal lobe which suggests that what we observe is probably a motor rhythm. The dataset under consideration did in fact contain a button press task which could explain the presence of such an atom.

Figure B.4: First 20 atoms learned on the MNE-somatosnesory dataset.

Figure B.5: Last 20 atoms learned on the MNE-somatosnesory dataset.

Figure B.6: A selection of A. temporal waveforms of the atoms learned on the MNE sample dataset, and their corresponding B. activations, and C. spatial patterns

Figure B.7: Dipole fit and power spectral density computed on MNE somato sample dataset for the atom in first row in Figure Figure B.6.