[Reviews · NeurIPS 2018]

Reviewer 1



This work extends convolutional sparse coding to the multivariate case with a focus on multichannel EEG decomposition. This corresponds to a non-convex minimization problem and a local minimum is found via an alternating optimization. Reasonable efficient bookkeeping (precomputation of certain factors, and windowing for locally greedy coordinate descent) is used to improve scalability. The locally greedy coordinate descent cycles time windows, but computes a greedy coordinate descent within each window. As spatial patterns are essential for understanding EEG, this multivariate extension is an important contribution. The work is clearly presented. The results demonstrate an efficient implementation. Some preliminary experiments show the potential for automatically learning brain waveforms. Weaknesses: In previous work, Jas et al. [13] consider different noise whereas this model assumes Gaussian noise versus the alpha stable noise. The authors should comment how this work could be extended to a different noise model. Parameter/model selection ($L$ and $K$) are not discussed for the real-world signal (lines 235–253). Choosing $K$ too large can lead to cases where some atoms are never used or updated, and other cases where the same waveform appears as different atoms (perhaps at different shifts). Perspectives on these real-world considerations should be mentioned. All the learned waveforms should be reported in the supplement. There are obvious similarities between greedy coordinate descent and the well-known matching pursuit algorithm, which is not mentioned. With the non-negative restriction, matching pursuit for time-series would have the same update in Eq. 5 without the shrinkage by lambda on the numerator. Assuming unit-norm patterns the greedy coordinate descent strategy (Eq. 7) would match the matching pursuit selection of maximal inner-product. Thus, the only change is that matching pursuit is done without shrinkage (lambda). Using matching pursuit, the remainder of the multivariate framework would not need to change. However, to be as efficient as the locally greedy version, a windowed version of matching pursuit should be done, which itself would be a novel algorithm. At least the issues with $\lambda_{max}$ could be replaced by using a fixed cardinality on the support. The authors could consider a discussion of the relation to matching pursuit as it has been used for both template matching (including the multivariate case) and learning waveforms in neural signals. Also to facilitate future comparisons the cardinality of support of the sparse code for different values of lambda could be investigated and reported. Relevant reference: Piotr J. Durka, Artur Matysiak, Eduardo Martínez Montes, Pedro Valdés Sosa, Katarzyna J. Blinowska, Multichannel matching pursuit and EEG inverse solutions, Journal of Neuroscience Methods, Volume 148, Issue 1, 2005, Pages 49-59. Based on my understanding of the reasoning, the authors believe the improved convergence efficiencies of a complete greedy coordinate descent would not compensate for the increased computational complexity of the complete greedy search. However, the gap between the local and globally greedy version could be narrowed: Couldn't the algorithm be improved by recomputing the differences in Eq. 7 after the update in each window, keeping them in a priority queue, and then proceeding to choose the best window? This could be implemented after the first loop through the $M$ windows in Algorithm 1. Lines 218–227 How are the spatial maps in this example created? From the text it appears they are distinct for the two atoms. But the multivariate approach still would be better than single channel even if the spatial maps were the same due to spatial independence of noise. This could be clarified. Figure 4. How was this atom selected (by manual inspection)? Is it representative or simply the most mu-like waveform? How often does this atom appear? Are there similar atoms to it with different spatial patterns? Minor issues: Line 20 "is central" is perhaps an overstatement in the context of neuroscience. "can be used to investigate" would be more reasonable. Line 22 duration -> durations Line 83 "aka" Line 98 overlapping -> overlapped Line 104 multivariate with -> multivariate CSC with Equations 2, 3 & 4 Should indicate the full sets of parameters as arguments in the minimizations. Especially to distinguish the case for a single multivariate signal in Equation 4. Line 246 "9 Hz.," Line 252 "In notably includes" Line 178 values -> value Line 179 "allows to use" Line 335 'et al.' in author list for reference [29] without additional authors. Line 356 'et al.' in author list for reference [38] with only one more author. ----- The authors have answered many of my questions. I look forward to a final version that will incorporate the additional discussion.

Reviewer 2



This paper presents an extension of convolutional sparse coding for electromagnetic brain signal analysis. The idea is to impose a rank-one constraint on every dictionary atom (channel x time), which is justified due to the instantaneous nature of the forward model. An efficient algorithm for estimating sparse codes is developed based on locally greedy coordinate descent (LGCD) method developed in a previous work. Experiment on real MEG data shows that the algorithm is more efficient than existing ones and the method can learn reasonable pairs of spatial and temporal atoms related to the mu-rhythm located at the somatosensory area. The use of rank-one constraint is well motivated for applications to MEG or EEG. Although technically rather straightforward, the combination with CSC in the particular application context sounds fairly an original idea and will be practically very useful. The application to real MEG data well demonstrates the validity. The presentation is clear, although the paper seems to put too much emphasis on algorithmic details rather than the data analysis results. About the timing comparison, the L-BFGS method by Jas et al. [13] appears to have a similar performance with the proposed method in the univariate case (Fig 1b). The improvement in large lambda is evident but the difference is relatively small as compared to the difference to the other methods. I suppose the L-BFGS method can readily apply to the multivariate or rank-one constraint cases by just replacing the Z-step, so the practical advantage of using LGCD may not be very large. The computational speed appears to be similar between the proposed multivariate and rank-one models, although I supposed rank-one constraint generally improves the speed by reducing the number of parameters. If it could be shown in some setting, this point may increase the relevance of the proposed rank-one model. About the MEG application, the figure only displays one atom obtained by the rank-one model. Since the rank-one constraint is the key idea of the proposed method, it should be compared with the multivariate model without the constraint (possibly with post-hoc rank-one approximation). Moreover, what was the nubmer of dictionary atoms in this experiment? It would be better if some comments on how other atoms look like as well as the robustness of the result shown here. Some symbols like [[1,T]] (=1,2,..,T) or that of the floor function are used without definition. It will be helpful for readers if they are provided in the text.

Reviewer 3



This paper proposes a method for a convolution sparse coding (CSC) for multi-variate signals. The problem is relaxed by imposing a rank-1 constraint on the dictionary and optimized by iterative steps to solve locally greedy coordinate descent. The method runs efficiently and is applied to characterize MEG signals from the brain which results in identifying different shaped signals in the same frequency bands. The paper is quite well written and here are some of my concerns. 1. The intuition why rank-1 constraint is used and why this would yield a good result is not well explained. 2. Label for y-axis in Fig. 1 a) and c) are missing. 3. There needs to be more description on datasets in the experiment. E.g., what is MEG data and why is it measured? What is the sample size? 4. Does the findings in the experiment have any relationship with existing literature in neuroscience? Is there any clinical justification or interpretation?